# Evaluating topological variability in Neodermata phylogenies using mitochondrial and ribosomal gene markers

Víctor Hugo Caña-Bozada[1,2*], Geormery Belén Mera-Loor[3],
David I. Hernández-Mena[4*], Jean-Lou Justine[5], Marta Álvarez-Presas[6]

1 Centro de Investigación para la Salud en América Latina (CISeAL), Pontificia Universidad Católica del Ecuador (PUCE), Quito, Ecuador, 2 Centro de Investigación en Alimentación y Desarrollo, A.C. Unidad Mazatlán en Acuicultura y Manejo Ambiental, Mazatlán, Sinaloa, Mexico, 3 Secretaría de Educación Superior, Ciencia, Tecnología e Innovación. Instituto Superior Tecnológico Luis Arboleda Martínez. Carrera de Tecnología Superior en Acuicultura, Ext. Jaramijó. Manabi, Ecuador, 4 Colección Nacional de Helmintos, Instituto de Biología, Universidad Nacional Autónoma de México, Ciudad Universitaria, Avenida Universidad número, Ciudad de México, Mexico, 5 ISYEB, Institut de Systématique, Évolution, Biodiversité (UMR7205 CNRS, EPHE, MNHN, UPMC, Université des Antilles), Muséum National d'Histoire Naturelle, Paris Cedex, France, 6 Institut de Biologia Evolutiva (CSIC-Universitat Pompeu Fabra), Passeig Marítim de la Barceloneta, Barcelona, Spain

* vhcana@puce.edu.ec (VHCB); dhernandez@ib.unam.mx (DIHM)

## Abstract

The Neodermata is a group of parasitic flatworms that includes the classes Trematoda, Cestoda, and Monogenea. Understanding the phylogenetic relationships within the Neodermata has been a longstanding challenge. Molecular studies utilizing different datasets have produced variable results, leading to differing evolutionary hypotheses. Resolving the phylogenetic relationships requires careful consideration of the molecular targets and sequences used. In this study, our objective was to investigate the topological variability of phylogenetic trees by examining different mitochondrial genes, molecular datasets (nucleotides and amino acids), as well as the 18S and 28S nuclear rRNA genes, and three software packages used for phylogenetic analysis. To evaluate the utility of different markers, we constructed 96 unilocus trees and nine multilocus trees. Our findings revealed that each gene provided unique information and resulted in different topologies depending on the sequences used, with only few mitochondrial genes indicating the monophyly of the Monogenea. Multilocus analyses mitochondrial and mitochondrial + 18S + 28S produced a consistent topology, supporting the monophyly of each of the four major neodermatan lineages (Cestoda, Trematoda, Monopisthocotylea, and Polyopisthocotylea). Notably, the monophyly of the Polyopisthocotylea and Cestoda consistently appeared in the different analyses. Conversely, we observed discrepancies between results obtained from mitochondrial genes and nuclear genes. This study contributes to our understanding of the phylogeny of the Neodermata by examining the topological variability of phylogenetic trees using both mitochondrial and nuclear genes. Our results emphasize

**Data availability statement:** All data are available in public databases and as supplementary file (S1 Data; https://github.com/victorcana/Topological_variability_Neodermata). Accession number is available in Table 1 of the manuscript.

**Funding:** The authors report the following sources of funding: ISTLAM (ISTLAM-OCS-RES.2023-288), awarded to G.B.M.L., and PAPIIT-UNAM (IA206125), awarded to D.I.H.M.

**Competing interests:** The authors have declared that no competing interests exist.

that carefully selected molecular markers and multilocus approaches are crucial for illuminating the complex evolutionary history within the Neodermata.

## Introduction

The Neodermata [1] is a diverse group of parasitic platyhelminths encompassing the classes Trematoda, Cestoda, and Monogenea. The Monogenea is divided into the subclasses Monopisthocotylea and Polyopisthocotylea; phylogenetic hypotheses with genetic data, and some with morphological data, indicate that these two subclasses are not nested in a monophyletic group [2–5]. The Neodermata comprises species with varied life histories, including both ecto- and endoparasites, as well as parasites with complex and simple life cycles. Molecular and morphological studies have established the Neodermata as a monophyletic group. Among the most notable morphological characters is the presence of a neodermis and the process of spermiogenesis [1,6], which provide a solid basis for considering the Neodermata as a monophyletic group. Despite this clear monophyly, unraveling the phylogenetic relationships between the different taxonomic classes and subclasses within the Neodermata has proven to be a challenge, mainly due to the inconsistencies observed in the results of different morphological and molecular studies of higher taxonomic levels such as subclass and class [5,7,8] (see Fig 1A-1G for details). Despite significant advancements in mitochondrial genome sequencing, substantial discrepancies persist between phylogenetic relationships inferred from mitochondrial data and those based on other molecular datasets, including nuclear genes, genomes, and transcriptomes.

Among the commonly used molecular markers are mitochondrial genes, which have become widely popular for barcoding studies, i.e., species-level identification and classification, due to their wide availability across many taxa and their rapid evolutionary rate. However, mitochondrial genes have also played a key role in phylogenetic studies, revealing several hypotheses about phylogenetic relationships within the Neodermata: Monopisthocotylea + (Polyopisthocotylea + (Trematoda + Cestoda)) [12], and Polyopisthocotylea + (Monopisthocotylea + (Trematoda + Cestoda) [8]. However, multilocus analysis using nuclear genes [5,17] and some unilocus analysis using 18S and 28S rRNA genes [7,18] do not support some hypotheses, including the monophyly of Trematoda + Cestoda (Fig 1A-1G).

It is important to mention that in the case of mitochondrial genomes, analysis of sequences at the nucleotide and amino acid level makes it possible to capture different aspects of molecular evolution, such as nucleotide substitutions and amino acid changes [19]. This dual perspective can help disentangle phylogenetic signal from noise and could explain some of the topological variability observed in previous studies. However, a comparative assessment of topological variability derived from different mitochondrial genes and inference methods is still lacking. Furthermore, although single-locus data are often considered limited for resolving deep phylogenies, they are still frequently used in taxonomic and systematic studies of parasitic platyhelminthes, especially when multilocus datasets are unavailable [20,21]. Therefore, an

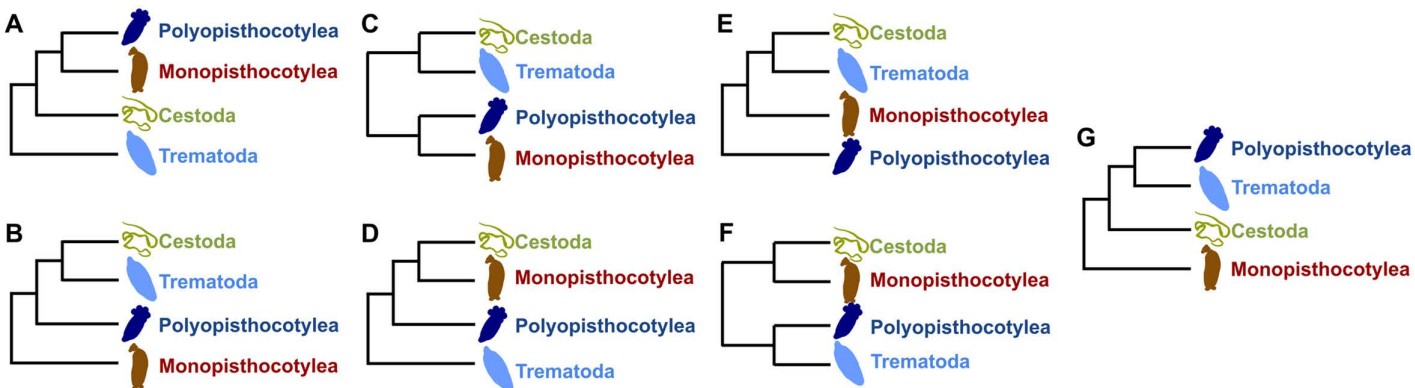

**Fig 1. Phylogenetic hypotheses of Neodermata.** Summary of the phylogenetic relationships of the four major neodermatan lineages proposed in previous studies based on morphological and molecular data. Topology obtained from data: **A)** Morphological [1,9], 18S rDNA [7], morphological + 18S rDNA; **B)** 18S rDNA [10], 28S rDNA [11], mitochondrial genome [12]; **C)** 28S rDNA, 18S + 28S rDNA [10]; **D)** 18S rDNA [13], 16S MrRNA + 18S NrRNA + 28S NrRNA + *cytb* [14]; E) mitochondrial genome [8,15]; F) nuclear genome [5]; G) nuclear genome [15,16].

updated and critical assessment of their phylogenetic behavior remains relevant, especially since these markers continue to be used in a wide range of evolutionary and diagnostic contexts.

Thus, the goal of this study was to investigate the variability of phylogenetic tree topologies within the Neodermata using different mitochondrial genes and molecular datasets (nucleotides and amino acids), as well as the 18S and 28S nuclear rRNA genes, employing three software packages and two different inference methods (Maximum Likelihood and Bayesian Inference) for phylogenetic analysis. We focused on evaluating mainly the groups: 1) Trematoda + Cestoda, 2) Trematoda + Monopisthocotylea, 3) Trematoda + Polyopisthocotylea, 4) Cestoda + Monopisthocotylea, 5) Cestoda + Polyopisthocotylea and 6) Monopisthocotylea + Polyopisthocotylea; additionally, we also evaluate the monophyly of each of the major neodermatan lineages: 7) Monopisthocotylea, 8) Polyopisthocotylea, 9) Trematoda, and 10) Cestoda. By incorporating a larger dataset and implementing multi-gene comparative analysis and phylogenetic inference methods, this study expands the scope of previous studies with similar approach [14,21].

## Materials and methods

### Genetic dataset

To explore the topological variability obtained according to different molecular markers and software used in phylogenetic analysis, unilocus and multilocus trees were constructed using the nucleotide and amino acid sequences from the mitochondrial genes of 16 neodermatans downloaded from GenBank (Table 1). Representative species were chosen from each major neodermatan lineage to ensure a diverse sample reflecting variability within this taxon. For tree rooting in the phylogenetic analyses, we used sequences from two planarian species—*Schmidtea mediterranea* and *Macrostomum lignano*—which are closely related free-living flatworms. This choice is consistent with previous studies (e.g., Caña-Bozada et al. [5]) and helps minimize long-branch attraction and rooting artifacts. In addition, the topological variability was also evaluated using the nuclear 28S and 18S rRNA sequences for each species. The data, alignments and analysis code is publicly available at https://github.com/victorcana/Topological_variability_Neodermata.

### Unilocus phylogenetic analyses

Unilocus phylogenetic analyses were performed for each Mitochondrial Protein-Coding Gene (MPCG), mitochondrial rRNA gene (MrRNA), and nuclear rRNA gene (NrRNA). Codon-based alignment was performed for the nucleotide

**Table 1. Species used for comparative phylogenetic analyses and GenBank accession numbers.**

| Species | Class | Subclass | Superfamily | Family | 28S ID | 18S ID | Mitochondrial Genome ID |
|---|---|---|---|---|---|---|---|
| *Echinococcus multilocularis* | Cestoda | Eucestoda | Cyclophyllidea | Taeniidae | pathogen_EMU_contig_0381 | AB731634.1 | AB018440.2 |
| *Hymenolepis microstoma* | Cestoda | Eucestoda | Cyclophyllidea | Hymenolepididae | LC064144.1 | AJ287525.1 | LC102493.1 |
| *Schistocephalus solidus* | Cestoda | Eucestoda | Diphyllobothriidea | Diphyllobothriidae | KY552833.1 | AF124460.1 | AP017669.1 |
| *Taenia asiatica* | Cestoda | Eucestoda | Cyclophyllidea | Taeniidae | AF004720.1 | GQ260088.1 | AP017670.1 |
| *Gyrodactylus salaris* | Monogenea | Monopisthocotylea | Gyrodactylidea | Gyrodactylidae | FJ971996.1 | Z26942.1 | NC_008815.1 |
| *Neobenedenia melleni* | Monogenea | Monopisthocotylea | Capsalidea | Capsalidae | EU707805.1 | EU707804.1 | JQ038228.1 |
| *Rhabdosynochus viridisi* | Monogenea | Monopisthocotylea | Dactylogyridea | Diplectanidae | TRINITY_DN1459_c0_g1_i5* | TRINITY_DN1459_c0_g1_i5* | MW565922.1 |
| *Scutogyrus longicornis* | Monogenea | Monopisthocotylea | Dactylogyridea | Ancyrocephalidae | HQ010035.1 | TRINITY_DN389_c0_g1_i1* | NC_056186.1 |
| *Eudiplozoon nipponicum* | Monogenea | Polyopisthocotylea | Mazocraeidea | Diplozoidae | AF382037.1 | AJ287510.1 | MW704020.1 |
| *Heterobothrium okamotoi* | Monogenea | Polyopisthocotylea | Mazocraeidea | Diclidophoridae | LC658930.1 | AB162155.1 | MK948930.1 |
| *Microcotyle sebastis* | Monogenea | Polyopisthocotylea | Mazocraeidea | Microcotylidae | AF382051.1 | AJ287540.1 | DQ412044.1 |
| *Paradiplozoon opsariichthydis* | Monogenea | Polyopisthocotylea | Mazocraeidea | Diplozoidae | KT781100.1** | KY640614.1*** | MG458327.1 |
| *Clonorchis sinensis* | Trematoda | Digenea | Opisthorchiida | Opisthorchiidae | 3WBP | scf00016 | MT607652.1 |
| *Fasciola hepatica* | Trematoda | Digenea | Plagiorchiida | Fasciolidae | JQ999969.1 | ON661086.1 | AF216697.1 |
| *Schistosoma mansoni* | Trematoda | Digenea | Strigeidida | Schistosomatidae | AY157173.1 | U65657.1 | AF216698.1 |
| *Trichobilharzia regenti* | Trematoda | Digenea | Strigeidida | Schistosomatidae | AY157244.1 | AY157218.1 | NC_009680.1 |
| *Macrostomum lignano* | Rhabditophora | N/A | Macrostomida | Macrostomidae | FJ715326.1 | FJ715306.1 | NC_035255.1 |
| *Schmidtea mediterranea* | Rhabditophora | N/A | Tricladida | Dugesiidae | DQ665992.1**** | AF013152.1 | JX398125.1 |

* 28S and 18S sequences obtained from Caña-Bozada et al. [6]; ** 18S sequence belongs to Paradiplozoon hemiculteri; *** 18S sequence belongs to Paradiplozoon yunnanensis; **** 18S sequence belongs to Schmidtea polychroa.

sequences of each MPCG with MACSE v.2 [22]. The amino acid sequences of each MPCG and the nucleotide sequences of each rRNA gene (12S and 16S MrRNAs; 18S and 28S NrRNAs) were aligned with MAFFT v7.31 (with the option --auto) [23]. Then, the gaps of all aligned sequences were trimmed with TrimAl [24], using the automated mode (-automated1). These programs were run from the graphical software PhyloSuite [25]. The best evolutionary model for each alignment was inferred with the ModelFinder program [26], based on the corrected Akaike information criterion. For mitochondrial aminoacid alignments mitochondrial-specific models such as mtZOA and mtInv were selected as the best fit to most alignments (e.g., atp6, cox1, nad1 with mtZOA; cox2, nad6 with mtInv). For mitochondrial nucleotide alignments, IQtre selected models such as GTR, TVM, and TIM, which are commonly used in mitochondrial phylogenetics due to their ability to accommodate compositional biases and rate heterogeneity. The best models were used to infer Maximum Likelihood (ML) phylogenetic trees with RAxML v8 [27] (1000 bootstrap iterations), and IQ-TREE v1.6.12 [28] using the

Shimodaira–Hasegawa-like approximate likelihood ratio test (SH-aLRT) (1000 replicates). Bayesian inference (BI) was performed using MrBayes v.3.2.7 [29] over two million generations, sampling the Markov chain at a frequency of 100 generations and using the default settings. The trees were visualized with FigTree v1.4.2 (http://tree.bio.ed.ac.uk/software/figtree/).

## Multilocus phylogenetic analysis

Multilocus phylogenetic analyses were performed on three gene sets: 1) nucleotide sequences of 12 MPCG (MPCG_NT) and 2 MrRNAs; 2) amino acid sequences of 12 MPCG (MPCG_AA) (Mito_AA); and 3) nucleotide sequences of 12 MPCG_NT + 2 MrRNA + 2 NrRNA. The previously aligned and trimmed sequences of each gene were concatenated using PhyloSuite for the construction of multilocus phylogenetic trees. PartitionFinder 2 [30] was used to select the optimal partition scheme and evolutionary model for each partition based on the corrected Akaike information criterion (AICc). The tree was constructed and visualized using the methodology described in Section "Unilocus phylogenetic analyses". Additionally, we conducted multilocus phylogenetic analyses using the same genes employed by Laumer and Giribet [14] (16S MrRNAs + 18S NrRNA + 28S NrRNA + *cytb*) to compare the resulting topologies.

## Phylogenetic distance analysis

To explore the phylogenetic distance in neodermatan species, the ML distance matrix of each gene was retrieved from IQ-TREE. IQ-TREE calculates the pairwise ML distances based on the estimated model parameters of the input sequences. Significance values were obtained by applying the Kruskal-Wallis statistical tests and the Dunn pairwise comparison tests. Significant differences and graphs were generated using the R package 'ggstatplot' [31].

## Topological comparisons: Robinson-Fould's distance

Robinson-Fould's distance (RF) was used to examine heterogeneity between each phylogenetic tree [32]. The RF distance was calculated using IQ-TREE. Since the distance analysis in IQ-TREE only accepts Newick format files as input, when necessary, the trees obtained in Nexus format were transformed using the 'Gotree' package [33]. K-means clustering based on the RF distance was performed using the function kmeans from R package 'stats' v4.3.2. The functions fviz_nbclust and fviz_cluster from the R package 'factoextra' v1.0.7 were used to determine the optimal number of clusters and visualize, in a two-dimensional space, clustering results [34]. Pairwise RF distances were visualized by generating a heatmap using the 'fviz_dist' function from the R package 'factoextra'.

## Saturation tests

Genes contain positions that undergo different substitution rate degrees due to evolutionary trajectories, which can contribute to the disparities in phylogenetic inference due to underestimating genetic distances among taxa. To explore this phenomenon, saturation levels were assessed in the sequence alignments of each MPCG_NT, MPCG_AA, MrRNA, and NrRNA using the method of comparison of patristic and uncorrected distances [35], following the approach outlined by Philippe *et al.* [36], utilizing PhyKIT [37]. This analysis evaluates the saturation level by comparing the number of substitutions inferred from the ML tree with the number of differences observed for each pair of species, derived from the complete alignment, where the increase in observed substitutions relative to inferred substitutions will indicate a higher saturation levels. In this analysis, values of 1 indicate no saturation, and values of 0 indicate complete saturation.

Furthermore, saturation levels were evaluated in each MPCG_NT after systematically excluding the first, second, and third codon positions from the alignment. The removal of these positions was performed using trimAl. Due to the higher saturation level observed in the third codon position, 12 phylogenetic trees (one for each MPCG_NT) were constructed using the sequence alignment devoid of this codon position. IQ-TREE was employed for tree construction, following the methodology described earlier.

## Results

### Phylogenetic trees

To assess their utility as phylogenetic molecular markers, we constructed 90 unilocus trees and six multilocus trees using nucleotide and amino acid sequences derived from mitochondrial genes of 16 neodermatan species. These analyses were conducted employing three different programs (S1 Fig). In addition, three multilocus trees were inferred using nucleotide sequences of mitochondrial genes + 18S NrRNA + 28S NrRNA, three multilocus trees using nucleotide sequences of 16S MrRNA + 18S NrRNA + 28S NrRNA + *cytb* (similar to Laumer and Giribet [14]), and six unilocus trees using NrRNA (S1 Fig). The best partition scheme and the optimal models of molecular evolution for the phylogenetic analyses are presented in S1 Table and S2 Table. The topology and phylogenetic distance varied depending on the gene, gene sets, and molecular datasets used in the phylogenetic analyses (Table 2 and S1 Fig). However, the topologies of the phylogenetic trees obtained with the three analysis software were generally similar for each gene and for each set of genes. The lengths of the final matrices for each analysis are presented in S3 Table.

Regarding the groupings of the four major lineages of the Neodermata that were evaluated in this study, the clades that appeared most times (in unilocus and multilocus trees), without considering the statistical support values, were the Cestoda + Monopisthocotylea (observed in 22 trees) and the Trematoda + Cestoda (observed in 18 trees). The Monopisthocotylea + Polyopisthocotylea clade (Monophyly of the Monogenea) was found in only 10 trees (Table 2 and S4 Table). In the multilocus mitochondrial trees, only the monophyly of the Trematoda + Cestoda clade was observed, while the multilocus mitochondrial + NrRNA phylogenetic analyses only recovered the monophyly of the Cestoda + Monopisthocotylea.

On the other hand, regarding the monophyly evaluated of each major lineage of the Neodermata, the Polyopisthocotylea and Cestoda were the most common monophyletic groups, appearing in 104 and 101 trees respectively (Table 2 and S4 Table). Only the multilocus mitochondrial (nucleotides and amino acids), multilocus mitochondrial (nucleotides) + 18S NrRNA + 28S NrRNA, *atp6* (nucleotides), multilocus 16S MrRNA + 18S NrRNA + 28S NrRNA + *cytb*, *cox1* (amino acids),

**Table 2. Number of topologies found using different datasets and software.**

| Software | Dataset | Trem+Cest | | Trem+Mono | | Trem+Poly | | Cest+Mono | | Cest+Poly | | Mono+Poly | | Mono-phyly Mono | | Mono-phyly Poly | | Mono-phyly Trem | | Mono-phyly Cest | |
|---|---|---|---|---|---|---|---|---|---|---|---|---|---|---|---|---|---|---|---|---|---|
| | | AA | NT | AA | NT | AA | NT | AA | NT | AA | NT | AA | NT | AA | NT | AA | NT | AA | NT | AA | NT |
| RAXML | Unilocus mitochondrial | 3 | 1 | 0 | 1 | 0 | 1 | 1 | 3 | 0 | 0 | 1 | 3 | 6 | 8 | 12 | 14 | 1 | 1 | 12 | 13 |
| RAXML | Multilocus mitochondrial | 1 | 1 | 0 | 0 | 0 | 0 | 0 | 0 | 0 | 0 | 0 | 0 | 1 | 1 | 1 | 1 | 1 | 1 | 1 | 1 |
| RAXML | Multilocus mitochondrial + 18S + 28S | 0 | 0 | 0 | 0 | 0 | 0 | 0 | 1 | 0 | 0 | 0 | 0 | 0 | 1 | 0 | 1 | 0 | 1 | 0 | 1 |
| RAXML | Unilocus 18S and 28S rRNA | 0 | 0 | 0 | 0 | 0 | 0 | 0 | 1 | 0 | 1 | 0 | 0 | 0 | 2 | 0 | 2 | 0 | 2 | 0 | 2 |
| IQTREE | Unilocus mitochondrial | 3 | 1 | 0 | 1 | 0 | 1 | 1 | 4 | 0 | 0 | 1 | 2 | 6 | 9 | 12 | 14 | 1 | 3 | 12 | 13 |
| IQTREE | Multilocus mitochondrial | 1 | 1 | 0 | 0 | 0 | 0 | 0 | 0 | 0 | 0 | 0 | 0 | 1 | 1 | 1 | 1 | 1 | 1 | 1 | 1 |
| IQTREE | Multilocus mitochondrial + 18S + 28S | 0 | 0 | 0 | 0 | 0 | 0 | 0 | 1 | 0 | 0 | 0 | 0 | 0 | 1 | 0 | 1 | 0 | 1 | 0 | 1 |
| IQTREE | Unilocus 18S and 28S rRNA | 0 | 0 | 0 | 0 | 0 | 0 | 0 | 1 | 0 | 1 | 0 | 0 | 0 | 2 | 0 | 2 | 0 | 2 | 0 | 2 |
| IQTREE | Unilocus mitochondrial codon 1 and 2 | 0 | 1 | 0 | 1 | 0 | 1 | 0 | 2 | 0 | 0 | 0 | 0 | 0 | 6 | 0 | 12 | 0 | 2 | 0 | 11 |
| MrBayes | Unilocus mitochondrial | 3 | 1 | 0 | 0 | 0 | 1 | 2 | 3 | 0 | 0 | 0 | 3 | 5 | 6 | 11 | 14 | 3 | 2 | 12 | 13 |
| MrBayes | Multilocus mitochondrial | 0 | 1 | 0 | 0 | 1 | 0 | 0 | 0 | 0 | 0 | 0 | 0 | 1 | 1 | 1 | 1 | 1 | 1 | 1 | 1 |
| MrBayes | Multilocus mitochondrial + 18S + 28S | 0 | 0 | 0 | 0 | 0 | 0 | 0 | 1 | 0 | 0 | 0 | 0 | 0 | 1 | 0 | 1 | 0 | 1 | 0 | 1 |
| MrBayes | Unilocus 18S and 28S rRNA | 0 | 0 | 0 | 0 | 0 | 0 | 0 | 1 | 0 | 1 | 0 | 0 | 0 | 2 | 0 | 2 | 0 | 2 | 0 | 2 |
| Total (39 trees of amino acids and 66 trees of nucleotides) | | 11 | 7 | 0 | 3 | 1 | 4 | 4 | 18 | 0 | 3 | 2 | 8 | 20 | 41 | 38 | 66 | 8 | 20 | 39 | 62 |

Trem, Trematoda; Cest, Cestoda; Mono, Monopisthocotylea; Poly, Polyopisthocotylea; AA, amino acid; NT, nucleotide.

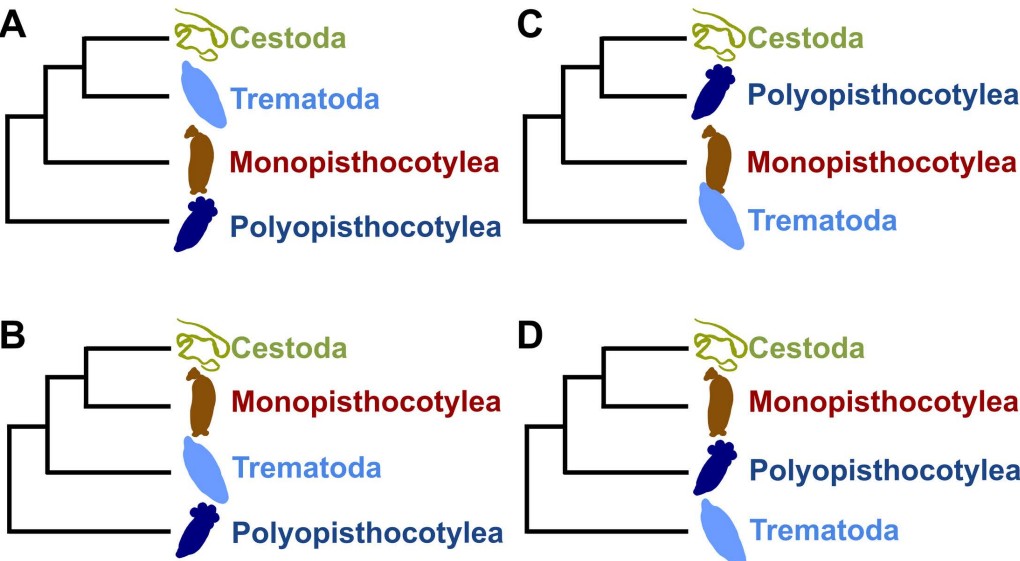

**Fig 2. Phylogenetic trees constructed from different datasets.** Phylogenetic trees indicate the monophyly of each of the four major lineages of the Neodermata. Topologies were obtained from A) the *cox*1 gene (amino acids) and multilocus mitochondrial (amino acids and nucleotides); B) multilocus 16S MrRNA + 18S NrRNA + 28S NrRNA + *cytb* (nucleotides), multilocus mitochondrial (nucleotides) + 18S NrRNA + 28S NrRNA, *atp*6 gene (nucleotides); **C)** 18S NrRNA; **D)** 28S NrRNA.

18S NrRNA, and 28S NrRNA trees recovered the monophyly of each of the four major lineages of the Neodermata (Monopisthocotylea, Polyopisthocotylea, Trematoda, and Cestoda), although with different topologies. (Fig 2A-2D; S1 Fig). S4 Table summarizes the results corresponding to the 10 topologies evaluated in this study.

## Phylogenetic distance

At the level of nucleotides, the mitochondrial genes *cox*1, 12S MrRNA, and 16S MrRNA generated the lowest phylogenetic distance between neodermatan species, while the *nad*2, *nad*4, *nad*4L, and *nad*6 genes have the highest phylogenetic distance (p-value < 0.05) (Fig 3A; S5 Table). The phylogenetic distance obtained from the multilocus phylogenetic analyses was higher than that obtained from the unilocus (p-value < 0.05) (Fig 3A). The phylogenetic distance obtained from 18S and 28S NrRNA was lower than that obtained from mitochondrial genes (p-value < 0.05).

At the level of amino acid, the mitochondrial gene that generated the lowest phylogenetic distance was *cox*1, while the *atp*6, *cox*3, and *nad*2 genes have the highest phylogenetic distance (p-value < 0.05) (Fig 3B and S6 Table). The multilocus phylogenetic tree based on amino acid sequences generated a higher phylogenetic distance in comparison with unilocus trees (p-value < 0.05) (Fig 3B). All the p-values of the pairwise comparison of the phylogenetic distance from each dataset used are shown in S5 Table and S6 Table.

## Topological comparisons

The k-mean clustering analysis grouped the 105 topologies obtained from the phylogenetic analyses into 10 clusters (Fig 4 and S7 Table). In the k-means analysis conducted on RF distance, the first and second axes of k-means clustering explained 40.8% and 16.3% of the variation in tree topologies, respectively. Dim1 separated mainly multilocus analysis topologies from unilocus analysis topologies, while Dim2 mainly separated the *atp*6, *cytb*, and *nad*1 unilocus analysis topologies.

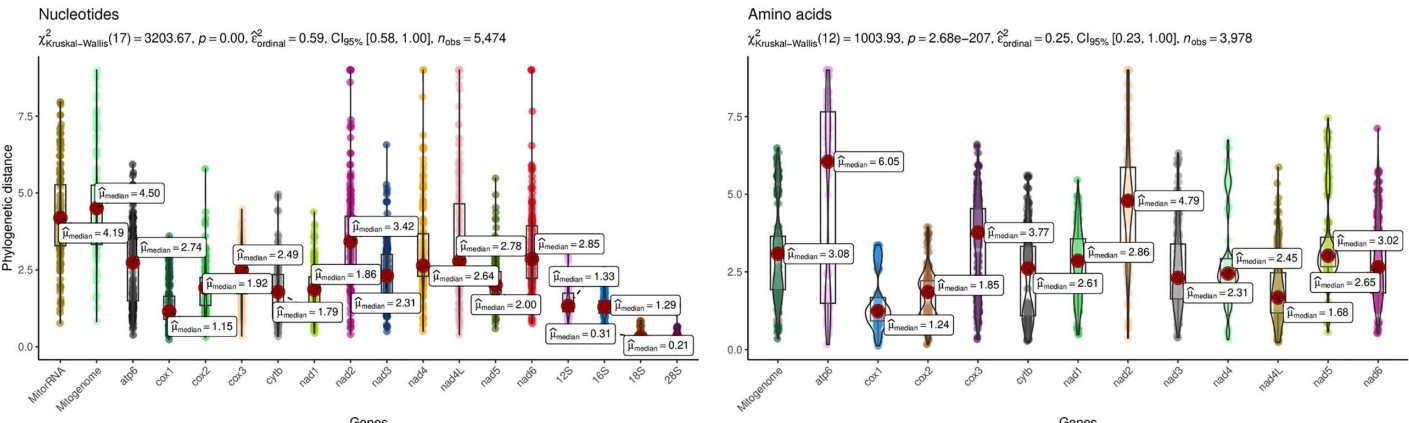

**Fig 3. Phylogenetic distance obtained from different datasets.** The datasets used included unilocus and multilocus data, as well as nucleotide and amino acid sequences. **A)** Phylogenetic distance obtained from nucleotide sequences. **B)** Phylogenetic distance obtained from amino acid sequences. P-values of the pairwise comparison of phylogenetic distance are shown in S5 Table and S6 Table. MitorRNA: multilocus trees constructed using sequences of mitochondrial genes + NrRNA (18S and 28S); Mitogenoma: multilocus trees constructed using sequences of mitochondrial genes.

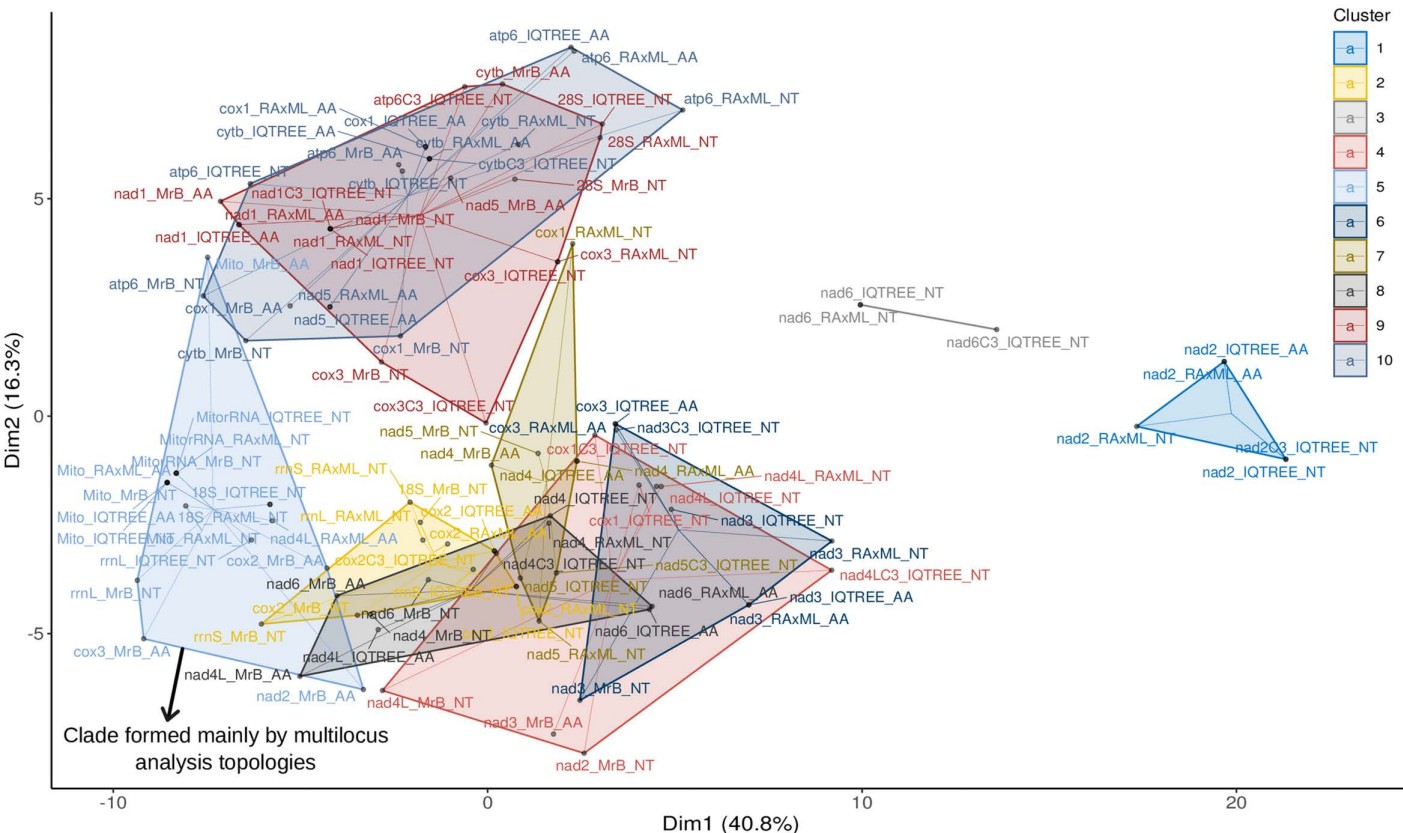

**Fig 4. Comparison of topologies among phylogenetic trees.** The scatterplot illustrates the outcomes of a k-means clustering analysis conducted on Robinson-Foulds (RF) distance matrices derived from 105 trees. The trees were constructed based on the mitochondrial, 18S NrRNA, and 28S NrRNA genes from 18 taxa and using different methods of phylogenetic inference (IQ-TREE, RAxML, and MrBayes). Each of the 10 clusters is shown in a different color. The dashed lines of each cluster converge at the centroid of the k-means analysis. NT, nucleotide sequence alignment; AA, amino acid sequence alignment; MrB, MrBayes; Mito, multilocus trees constructed using sequences of mitochondrial genes; MitorRNA, multilocus trees constructed using sequences of mitochondrial genes + NrRNA (18S and 28S).

## Saturation level in sequences

Nucleotide alignments, encompassing all three codon positions, exhibited significantly higher saturation levels compared to amino acids (S8 Table). After the alignment and gap trimming step, at the nucleotide level, genes such as 18S NrRNA, 28S NrRNA, *nad*2, *nad*4, *nad*4L, *nad*5, and *nad*6 showed the highest saturation levels, whereas, at the amino acid level, saturation was highest in *nad*2 and *nad*4. Interestingly, the gap-trimming step in the *atp*6 gene and each *nad* gene (except *nad*6) increased the saturation of the data, contrasting with the rest of the genes. By removing individual codon positions to investigate saturation effects on MPCG_NT, it became evident that the third codon exhibited the highest saturation. Despite this observation, eliminating this position did not lead to an improvement in tree topology (S1 Fig).

## Discussion

This study examines how different mitochondrial genes, molecular datasets (nucleotides and amino acids), and phylogenetic methods influence the reconstruction of relationships among the four main Neodermata lineages: Cestoda, Trematoda, Monopisthocotylea, and Polyopisthocotylea. Although multilocus and phylogenomic approaches are now the standard for deep phylogenetic inference, unilocus markers remain widely used in systematics, especially when genomic resources are limited. Therefore, our objective was to evaluate the consistency and limitations of these widely used markers when applied to high-level phylogenetic questions and to provide a comparative reference for interpreting topological variability. Specifically, we assessed the degree to which individual genes and combined datasets recover the monophyly of major Neodermata lineages and produce consistent topologies, providing insight into their utility and potential pitfalls in phylogenetic studies.

Individual utilization of each mitochondrial gene led to distinct topologies in the trees, with very few recovering the monophyly of each major lineage of the Neodermata. The most common groups obtained as monophyletic were the Polyopisthocotylea and Cestoda, and they were observed with the majority of the data sets, both individual genes and concatenated genes. On the other hand, few datasets recovered the monophyly of all major lineages of the Neodermata (between the Cestoda, Trematoda, Monopisthocotylea and Polyopisthocotylea): multilocus mitochondrial (nucleotides and amino acids), multilocus mitochondrial (nucleotides) + 18S NrRNA + 28S NrRNA, multilocus 16S MrRNA + 18S NrRNA + 28S NrRNA + *cytb*, *atp*6 (nucleotide), *cox*1 (amino acids), 18S NrRNA, and 28S NrRNA (Fig 2A-2D). It has been noted that mitochondrial genomes exhibit non-stationarity in nucleotide frequencies, with mitochondrial genome substitution rates in platyhelminths being over four times higher than those in other Bilateria, potentially providing suboptimal evidence at this resolution level [17,38]. The monophyly of the Monogenea was recovered by only a few mitochondrial genes, contradicting these genes the traditional classification of this class, both molecular and morphological data consistently indicate that the Monogenea does not form a monophyletic group [2,3,15].

The most corroborated relationships were the Cestoda + Monopisthocotylea and the Trematoda + Cestoda. The multilocus mitochondrial trees recovered the monophyly of the Trematoda + Cestoda, with the Monopisthocotylea positioned as their sister group (Fig 2A). This topology is consistent with the findings of Zhang *et al.* [8], using mitogenomic data. However, it is important to note a discrepancy observed in a previous mitogenomic study by Perkins *et al.* [12] (Fig 1), which placed the Monopisthocotylea as the sister group of a clade formed by the other three major lineages. This difference may be attributed to the use of only one polyopisthocotylean species in their analysis. In contrast, our study benefits from having more taxa considered in the analyses, allowing for a more comprehensive evaluation of phylogenetic relationships within the Neodermata.

On the other hand, the trees obtained from the data sets multilocus mitochondrial + 18S NrRNA + 28S NrRNA, multilocus 16S MrRNA + 18S NrRNA + 28S NrRNA + *cytb*, *atp*6 (nucleotides), and 28S NrRNA were those that recovered the monophyly of each major lineage of the Neodermata and the monophyly Monopisthocotylea + Cestoda (Fig 2B, D), consistent with findings from previous phylogenomic studies using nuclear genes ( [5,17]: Fig 1F). However, none of these trees were consistent with the Trematoda + Polyopisthocotylea clade previously reported by Caña-Bozada *et al.* [5]. This

discrepancy may be due to the non-stationarity of nucleotide frequencies or an accelerated mutation rate in mitochondrial genomes among platyhelminths [17,38,39], which diminish the power to resolve relationships among the four major lineages of the Neodermata.

The topology of major lineages observed in the 28S NrRNA trees of our study (Fig 2D) coincided with that obtained by Littlewood *et al.* [13] using 18S NrRNA and Laumer and Giribet [14] employing 16S MrRNA + 18S NrRNA + 28S NrRNA + *cytb*. However, previous studies using 28S NrRNA have reported different topologies ( [11]: Fig 1B; [10]: Fig 1C), none of which corresponded to ours. In the case of Mollaret *et al.* [11], this discrepancy may be attributed to the use of partial domains C1 and D2 along with the full domains D1 and C2 of the 28S ribosomal RNA, whereas, except for the 28S gene of *G. salaris*, we used the complete sequence. Conversely, the discrepancy with Lockyer *et al.* [10], which used the complete sequence, may be attributed to their including fewer representatives of the Monopisthocotylea and Polyopisthocotylea, with only two representatives of each.

We replicated the analysis of Laumer and Giribet [14] using the same genes but different species and numbers of taxa in each major neodermatan lineage to assess the influence of species or number of taxa. This resulted in a different topology in our study (Fig 1D and Fig 2B). Similar to Lockyer *et al.* [10], Laumer and Giribet [14] included only two representatives each of the Monopisthocotylea and Polyopisthocotylea, including the species *Udonella caligorum*. Although the number of representatives in the Monogenea can influence the topology, the discordance could also be related to the inclusion of species that generate long branches, such as *Udonella caligorum*. Notably, this species exhibits a unique biology as an epibiont on a copepod itself parasitic on a fish, distinguishing it from typical the Monopisthocotylea. According to morphological phylogenetic studies, *Udonella* spp. does not group with the Monogenea, reflecting the particularities of this species [13], emphasizing caution when including it in phylogenetic analyses, at least in unilocus analyses.

Additionally, our trees obtained from 18S NrRNA (Fig 2C) did not coincide with those of other studies using the same gene. This was observed in studies by Campos *et al.* [7] (Fig 1A), Littlewood *et al.* [13] (Fig 1D), and Lockyer *et al.* [10] (Fig 1B), all showing different topologies. These can be related to different attributes, such as the dataset used, software, or species. Campos *et al.* [7] used different programs with parameters different from ours, such as the use of a one-parameter maximum likelihood model with empirical base frequencies. Although the groups were well represented in Littlewood *et al.* [13], they utilized maximum parsimony and minimum evolution distance methods. While Lockyer *et al.* [10] used ML and BI methods and the complete sequence of 18S NrRNA, similar to our study, they included fewer representatives of monogeneans, with only two representatives of the Monopisthocotylea and Polyopisthocotylea.

The nucleotide sequences of *nad*2, *nad*4, *nad*4L, and *nad*6, as well as the amino acid sequences of *atp*6, *cox*3, and *nad*2, exhibited high phylogenetic distances. Their utilization led to trees with limited or no evidence for the monophyly of each major neodermatan lineage. Therefore, these genes are not recommended for phylogenetic studies of supraspecific groups such as the Neodermata classes or for species that are phylogenetically very distant because they can lead to erroneous topology. While the observed high divergence could be indicative of high substitution rates, which in other contexts have been associated with potential utility at shallow phylogenetic levels, assessing its performance for closely related species or population-level studies would require separate analysis and targeted sampling, which are beyond the scope of this study.

On the other hand, the nucleotide sequences of *cox*1, 12S MrRNA, and 16S MrRNA, as well as the amino acid sequence of *cox*1, generated the lowest phylogenetic distances. This is due to a relatively low mutation rate (non-synonymous and synonymous) compared to the other mitochondrial genes [39]. Despite the low phylogenetic distance obtained from the 12S MrRNA and 16S MrRNA genes, these were not informative for recovering the monophyly of the Trematoda, which does not make these markers suitable for exploring the phylogenetic relationships of supraspecific groups or of phylogenetically very distant species.

Despite efforts to reduce saturation in our data, overall improvements in tree topologies were not observed. The saturation level was higher in the nucleotide sequences than in the amino acid sequences, due to their smaller state space: while there are four possible bases for nucleotides, there are 20 possible amino acids for proteins [36]. The saturation

analysis was partially consistent with the results obtained regarding phylogenetic distance. Specifically, in both analyses, the mitochondrial genes *nad*2, *nad*4, *nad*4L, and *nad*6 at the nucleotide level, as well as *nad*2 at the amino acid level, exhibited the highest level of saturation and phylogenetic distance (S8 Table and Fig 3A, 3B). This contrasts with other groups of non-neodermatan platyhelminths, where *cox*1 is quite saturated [40]. Previous research indicates that *nad*2 is the second mitochondrial gene to accumulate the most non-synonymous mutations and is also the gene that accumulates the largest number of synonymous mutations within the Neodermata [40]. This finding may explain the elevated phylogenetic distance observed at both nucleotide and amino acid levels. In contrast, the *atp*6 and *cox*3 genes exhibit high values of phylogenetic distance at the amino acid level, despite not being the genes that accumulate the most non-synonymous mutations [39], a pattern that was not possible to explain with our analyses.

Other genes that exhibited high levels of saturation were the 18S NrRNA and 28S NrRNA genes, despite showing the lowest phylogenetic distance values. Similar to neodermatans, annelids also exhibit high levels of saturation in these genes [41], which increases the likelihood of artifacts resulting from the accumulation of multiple substitutions at the same position over time. While these artifacts may introduce inconsistencies in tree reconstruction [42], the inclusion of these genes in the phylogenetic analysis still led to the recovery of monophyly for each of the four main lineages of the Neodermata (Fig 2C, 2D). This can be attributed to the fact that these genes also contain highly conserved regions that remain unsaturated, and it is these regions that contribute to the recovery of monophyletic groups.

There is considerable inconsistency in the phylogenetic relationships inferred from different datasets, such as mitochondrial genes, nuclear genes, unilocus, or multilocus, which expands the discussion of phylogenetic relationships within and between the Neodermata through the multiple phylogenetic hypotheses that arise from their analysis. For instance, phylogenomic analyses have shown a sister relationship between the Trematoda and Polyopisthocotylea [5,16], mitogenomic analyses have indicated a sister relationship between the Trematoda and Cestoda [8], and 18S NrRNA analyses have depicted the Monogenea as a monophyletic group and others as non-monophyletic groups [7,10]. While our study supported the monophyly of each major lineage through mitochondrial multilocus analyses, this was not consistently observed in all cases using a similar dataset [12,43]. It is important to mention that, in addition to molecular markers and methods, taxon sampling is equally important. By expanding the number of representative species for each major clade of the Neodermata, phylogenetic studies can better capture phylogenetic relationships within and between these clades, improving the reliability and precision of phylogenetic inferences.

## Conclusions

In this study, we assessed the phylogenetic variability of mitochondrial and ribosomal genes in the Neodermata, focusing on the monophyly of major lineages using diverse molecular datasets and phylogenetic software. Our findings revealed that each mitochondrial gene provided different information and produced different topologies depending on the sequences used. Multilocus mitochondrial analyses, as well as those using unilocus 18S rRNA, 28S rRNA, and *cox*1 genes, produced a consistent topology and recovered the monophyly of each major neodermatan lineage (Trematoda, Cestoda, Polyopisthocotylea, Monopisthocotylea), although there was only minimal evidence for the monophyly of the Monogenea. On the other hand, mitochondrial genes such as *nad*2, *nad*4, and *nad*6 exhibit high phylogenetic variability and limited evidence for major lineage monophyly. These findings highlight the complexity involved in phylogenetic analyses within the Neodermata. While progress has been made, challenges remain, particularly related to the choice of molecular markers and the influence of taxonomic sampling on tree topology.

## Supporting information

**S1 Fig. Phylogenetic trees obtained from mitochondrial and nuclear genes of 16 species of the Neodermata.** Two species of planarians (*Schmidtea mediterranea* and *Macrostomum lignano*) were used as an outgroup, using three different software. Ninety unilocus trees and six multilocus trees were constructed using nucleotide and amino acid sequences

derived from mitochondrial genes. Furthermore, three multilocus trees were inferred using the nucleotide sequences of mitochondrial genes + 18S NrRNA + 28S NrRNA, three multilocus trees using the nucleotide sequences 16S MrRNA + 18S NrRNA + 28S NrRNA + cytb, and six unilocus trees using NrRNA.
(PDF)

**S1 Table. Selected evolutionary model for the construction of each unilocus phylogenetic tree.**
(XLSX)

**S2 Table. Partition and selected evolutionary model for the construction of unilocus phylogenetic trees.**
(XLSX)

**S3 Table. The lengths of the alignments used for each phylogenetic analysis and the number of species included.**
(XLSX)

**S4 Table. Number of topologies found using different datasets and software.** Trem, Trematoda; Cest, Cestoda; Mono, Monopisthocotylea; Poly, Polyopisthocotylea; AA, amino acid; NT, nucleotide. "-" indicates that no phylogenetic analyses were performed for that dataset.
(XLSX)

**S5 Table. P-values of the pairwise comparison of the phylogenetic distance from each dataset of nucleotides used.**
(XLSX)

**S6 Table. P-values of the pairwise comparison of the phylogenetic distance from each dataset of amino acids used.**
(XLSX)

**S7 Table. Comparison of topologies among phylogenetic trees using the Robinson-Foulds distances.** Robinson-Foulds distances were calculated for each pair of topologies among the 105 phylogenetic trees generated.
(XLSX)

**S8 Table. The saturation level estimated from various datasets of multiple sequence alignments, following the methodology outlined by Philippe et al. (2011).** AA, amino acids; NT, nucleotides.
(XLSX)

## Acknowledgments

Víctor Hugo Caña-Bozada thanks CONAHCYT for his graduate student scholarship.

## Author contributions

**Conceptualization:** Víctor Hugo Caña-Bozada, Jean-Lou Justine, Marta Álvarez-Presas.

**Data curation:** Víctor Hugo Caña-Bozada, Geormery Belén Mera-Loor.

**Formal analysis:** Víctor Hugo Caña-Bozada.

**Investigation:** Víctor Hugo Caña-Bozada, David Iván Hernández-Mena.

**Software:** Víctor Hugo Caña-Bozada.

**Supervision:** Víctor Hugo Caña-Bozada, Marta Álvarez-Presas.

**Visualization:** Víctor Hugo Caña-Bozada.

**Writing – original draft:** Víctor Hugo Caña-Bozada.

**Writing – review & editing:** Geormery Belén Mera-Loor, David Iván Hernández-Mena.

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
