## [Decision Letter · Decision Letter 0]

1 Jul 2025

PONE-D-25-05422Phylogenetic relationships among major clades of the Neodermata: Insights from mitochondrial and ribosomal gene analysesPLOS ONE

Dear Dr. Caña Bozada,

Thank you for submitting your manuscript to PLOS ONE. After careful consideration, we feel that it has merit but does not fully meet PLOS ONE’s publication criteria as it currently stands. Therefore, we invite you to submit a revised version of the manuscript that addresses the points raised during the review process. 

We look forward to receiving your revised manuscript.

Kind regards,

James Lee Crainey, Ph.D.

Academic Editor

PLOS ONE

Journal Requirements:

2. Please note that PLOS ONE has specific guidelines on code sharing for submissions in which author-generated code underpins the findings in the manuscript. In these cases, we expect all author-generated code to be made available without restrictions upon publication of the work. 

Please review our guidelines at https://journals.plos.org/plosone/s/materials-and-software-sharing#loc-sharing-code and ensure that your code is shared in a way that follows best practice and facilitates reproducibility and reuse.

“Víctor Hugo Caña-Bozada thanks CONAHCYT for his graduate student scholarship. This work was supported by the Instituto Superior Tecnológico Luis Arboleda Martínez through research projects: “Parásitos metazoarios en peces: implicaciones y control en la interacción patógeno-hospedero (ISTLAM-OCS-RES.2023-288)".

**Additional Editor Comments:**

If you choose to revise your manuscript for us, please make sure your revised work  conforms with the PLOS ONE publishing criteria detailed here: https://journals.plos.org/plosone/s/criteria-for-publication. However, please also remember that PLOS ONE evaluates articles based on “scientific validity, strong methodology, and high ethical standards—not perceived significance”. It is, thus, important that you focus your responses on the technical and methodological queries that the reviewers have raised. PLOS ONE reviewers are free to express their opinions all aspects of your work and we hope that all of their feedback is useful to you, but it is only necessary to address the methodological and technical concerns that they raise, not their concerns relating to the perceived importance of your work.

Reviewers' comments:

Reviewer's Responses to Questions

**Comments to the Author**

1. Is the manuscript technically sound, and do the data support the conclusions?

Reviewer #1: Yes

Reviewer #2: No

Reviewer #3: Partly

2. Has the statistical analysis been performed appropriately and rigorously? 

Reviewer #1: Yes

Reviewer #2: N/A

Reviewer #3: Yes

3. Have the authors made all data underlying the findings in their manuscript fully available?

Reviewer #1: Yes

Reviewer #2: Yes

Reviewer #3: Yes

4. Is the manuscript presented in an intelligible fashion and written in standard English?

Reviewer #1: Yes

Reviewer #2: Yes

Reviewer #3: Yes

5. Review Comments to the Author

Reviewer #1: Justification of responses and review comments:

The manuscript is technically sound, and the data support the conclusions presented. Phylogenetic analyses were conducted using a comprehensive dataset and well-established tools such as IQ-TREE, MrBayes, and RAxML. The statistical analysis was performed rigorously, with appropriate model selection and the use of support measures like bootstrap and Bayesian posterior probabilities.

The data used in the analyses are available for access, as stated in the manuscript, allowing verification of the information.

Overall, the manuscript is well written and clear, with detailed methodological descriptions and a logical flow of ideas. However, some improvements could strengthen clarity and scientific rigor. Specific suggestions include:

Lines 86–87: Remove the marked text. Mentions of software used should be limited to the methodology section.

Line 101: Although data availability is mentioned, it is recommended to add a more specific paragraph clearly indicating where the data, alignments, and scripts can be accessed.

Lines 106–109: The manuscript mentions only the species used as outgroups but does not justify this choice. It is important to explicitly explain the criteria used and consider including additional outgroup taxa to improve tree rooting.

Line 130: Although the use of ModelFinder is cited, there is no mention of whether the chosen models have been evaluated to be appropriate for mitochondrial data, which have particular evolutionary patterns. Please add this information.

Line 172: Removing the third codon position to address saturation is appropriate, but the use of heterogeneous models (e.g., CAT-GTR), which could enhance phylogenetic accuracy, is not discussed. Considering this approach is recommended.

In summary, this is a solid and relevant manuscript, but addressing these points will enhance clarity, reproducibility, and robustness of the analyses.

Reviewer #2: This is already my third opportunity to review this manuscript, for a third scientific journal. It’s important to note that despite the several important methodological limitations I’ve identified previously, the text of the manuscript remains virtually identical (it is only reformatted for another journal), and the authors seem to refuse putting any more work into the manuscript content. Therefore, and similar to my second review of this paper, I will only limit myself to pasting in my original evaluation, recommending the authors to reconsider the manuscript from the ground up. Please see my reasons bellow.

The importance of this manuscript is, in my opinion, very limited, given the following reasons: First, the authors did not generate any new original sequence data. Neither they tried to mine the wealth of available data generated under the recent transcriptomic projects (mainly the one by Brabec et al. (2023) The evolution of endoparasitism and complex life cycles in parasitic platyhelminths. Curr Biol 33:4269-4275.e3), to the exception of two transcriptomes (ie Rhabdosynochus, Scutogyrus) generated by themselves. They could have simply performed the same blast searches on the other transcriptome assemblies available in repositories. I'm sure they would be able to obtain a complete set of mt protein-coding genes and a nearly complete set (at minimum) of rRNA coding genes this way. Instead, they've missed a perfect opportunity to include many early branching representatives of the four major neodermatan lineages in their analyses. I don't think I need to emphasize the importance of the early-branching groups for understanding phylogenetic histories here.

Second, their selection and/or use of the outgroup taxa seems inappropriate given the availability of better alternatives (ie Bothrioplana, a well-sequenced representative of the closest known relative of the Neodermata) and their goal of evaluating the utility of the individual mt/rRNA single-gene markers, including their assessment of the relative genetic distances between taxa. The inclusion of an outgroup will inevitably affect branch lengths of the ingroup taxa. If the authors insisted on the use of an outgroup, I would probably suggest using some alternative approach, like the Evolutionary placement algorithm (doi.org/10.1093/sysbio/syr010) that has been implemented in RAxML. Even better, such an analysis might be better performed only including the ingroup taxa.

Third (and I confess this point might be rather subjective), I see their focus/topic of the paper somewhat exhausted and a thing of the past. Platyhelminth systematics have entered the era of phylogenomics a decade ago and the field seems now rather focused on determining novel (and genuine) phylogenetic signals from multiple genomic loci, identifying sources of phylogenetic noise and potential sources of systematic error in the data and tree inference methods. Here, two notoriously known genomic loci are being once again re-considered, both of which has been scrutinized number of times previously in a similar manner both independently and combined. The most recent paper on the topic I'm aware of was the one by Zhang et al (2024) Strong mitonuclear discordance in the phylogeny of Neodermata and evolutionary rates of Polyopisthocotylea. Int J Parasitol 54:213–223. But really, other numerous papers have already scrutinized the phylogenetic signal of mt vs nuclear rRNA (as well as within each locus) over the last twenty years both in the Neodermata and within individual major neodermatan groups. This point would be invalid if the manuscript brought some novel evidence (or data) but I don't see anything discussed in the text that I haven't seen before.

Additionally, I have been wondering about two other particular things but did not see a rationalization anywhere in the text: i) What was the particular motivation to analyse the notoriously known fast-evolving (ie mt protein-coding) genes at the nucleotide level when they have been shown to be saturated at significantly lower taxonomic levels previously in different groups of parasitic flatworms? ii) What was the motivation to employ two (largely substitute) maximum likelihood programs?

Based on those facts, I suggest the authors to reconsider the paper.

Reviewer #3: This study investigates the topological variability of phylogenies of the major Neodermata clades derived from using different genetic and molecular datasets. While this is an interesting exercise, I have some objections that should be considered before publication.

My main criticism concerns the use of unilocus data for phylogenetic inference involving high-order clades. For over two decades, it has been known that unilocus sequences are quite unreliable for this purpose (e.g. https://doi.org/10.1371/journal.pbio.1000602). So some conclusions (see 290-291) are trivial. I gather from the text that a possible reason could be to determine the genes useful for distinguishing between closely related species (92-93). However, this is totally outside the scope of the study and the subsequent discussion on the suitability of certain sequences to study intraspecific variation (354-369) is not directly backed by evidence from the study. Therefore, the authors should better explain the rationale and relevance for incorporating unilocus analyses in their study.

In a broader context, the anticipated relevance of the study is not sufficiently demonstrated in the Introduction. The study is largely based on reproducing an approach that was published over 10 years ago (Laumer and Giribert, 2014). Therefore, the current effort is not shown to be particularly innovative and the authors should provide a clearer explanation of the advantages of their approach and its potential contribution to the state of the art. Most evidence presented appears confirmatory of previous work.

Specific comments

The main goal of the study is to investigate topological variability in phylogenies resulting from the combination of different datasets. This is not adequately reflected in the title.

Fig. 2 does not seem to add anything additional to what has been indicated in the text.

252. “This analysis helped…” This sentence is not about a result, but an interpretation or explanation of results. Delete and elaborate in the discussion if required.

6. PLOS authors have the option to publish the peer review history of their article (what does this mean? ). If published, this will include your full peer review and any attached files.

**Do you want your identity to be public for this peer review?** For information about this choice, including consent withdrawal, please see our Privacy Policy .

Reviewer #1: No

Reviewer #2: No

Reviewer #3: No

---

## [Author Response · Author response to Decision Letter 1]

12 Aug 2025

Manuscript PONE-D-25-05422

Response to Reviewers' Comments

We thank the reviewers for their constructive comments, which we have addressed through changes to the manuscript. These changes are described in our detailed responses below and are marked in the manuscript.

Reviewer #1:

The manuscript is technically sound, and the data support the conclusions presented. Phylogenetic analyses were conducted using a comprehensive dataset and well-established tools such as IQ-TREE, MrBayes, and RAxML. The statistical analysis was performed rigorously, with appropriate model selection and the use of support measures like bootstrap and Bayesian posterior probabilities.

The data used in the analyses are available for access, as stated in the manuscript, allowing verification of the information.

Overall, the manuscript is well written and clear, with detailed methodological descriptions and a logical flow of ideas. However, some improvements could strengthen clarity and scientific rigor. Specific suggestions include:

Lines 86–87: Remove the marked text. Mentions of software used should be limited to the methodology section.

Reply: Done. We have removed the mention of the software outside the methods section as suggested.

Line 101: Although data availability is mentioned, it is recommended to add a more specific paragraph clearly indicating where the data, alignments, and scripts can be accessed.

Reply: Done. We have added details indicating where the data, alignments, and scripts can be accessed.

Lines 106–109: The manuscript mentions only the species used as outgroups but does not justify this choice. It is important to explicitly explain the criteria used and consider including additional outgroup taxa to improve tree rooting.

Reply: Done. We have added a justification for our choice of outgroups, noting that they were selected based on phylogenetic proximity to Neodermata and data completeness.

Line 130: Although the use of ModelFinder is cited, there is no mention of whether the chosen models have been evaluated to be appropriate for mitochondrial data, which have particular evolutionary patterns. Please add this information.

Reply: Done. An explanation of the selected model is now included in the main text.

Line 172: Removing the third codon position to address saturation is appropriate, but the use of heterogeneous models (e.g., CAT-GTR), which could enhance phylogenetic accuracy, is not discussed. Considering this approach is recommended.

Reply: We appreciate the reviewer's suggestion and agree that this type of model may be more appropriate. However, since these models are not available in programs such as IQ-tree or Raxml, they were not considered. Instead, we addressed site heterogeneity using models available for the programs used. However, we primarily applied the GTR+F+G4 model, selected by Modelfinder, and in one specific case, the FreeRate model (GTR+F+R4). Although CAT-GTR was not used directly, our strategy incorporated models including +G4 and +I+G4, and +R4, which efficiently account for site heterogeneity.

In summary, this is a solid and relevant manuscript, but addressing these points will enhance clarity, reproducibility, and robustness of the analyses.

Reviewer #2: This is already my third opportunity to review this manuscript, for a third scientific journal. It’s important to note that despite the several important methodological limitations I’ve identified previously, the text of the manuscript remains virtually identical (it is only reformatted for another journal), and the authors seem to refuse putting any more work into the manuscript content. Therefore, and similar to my second review of this paper, I will only limit myself to pasting in my original evaluation, recommending the authors to reconsider the manuscript from the ground up. Please see my reasons bellow.

The importance of this manuscript is, in my opinion, very limited, given the following reasons: First, the authors did not generate any new original sequence data.

Neither they tried to mine the wealth of available data generated under the recent transcriptomic projects (mainly the one by Brabec et al. (2023) The evolution of endoparasitism and complex life cycles in parasitic platyhelminths. Curr Biol 33:4269-4275.e3), to the exception of two transcriptomes (ie Rhabdosynochus, Scutogyrus) generated by themselves. They could have simply performed the same blast searches on the other transcriptome assemblies available in repositories. I'm sure they would be able to obtain a complete set of mt protein-coding genes and a nearly complete set (at minimum) of rRNA coding genes this way. Instead, they've missed a perfect opportunity to include many early branching representatives of the four major neodermatan lineages in their analyses. I don't think I need to emphasize the importance of the early-branching groups for understanding phylogenetic histories here.

Reply: We appreciate the reviewer's comments. This study is not intended to propose a new phylogenetic hypothesis for Neodermata, but rather to assess the consistency of mitochondrial and ribosomal markers commonly used in parasitological systematics. The objective of this study is to investigate the derivation of different phylogenetic hypotheses for the major groups of Neodermata (which have already been proposed elsewhere) given different molecular markers (mitochondrial and nuclear), different data sets (unilocus and multilocus), different types of molecules (nucleotides and amino acids), and different phylogenetic analysis programs. Of course, exhaustive taxonomic sampling would be critical and necessary only if one sought to explore phylogenetic relationships within each of the major clades (i.e., relationships within families, superfamilies, or orders of Trematoda, Cestoda, and Monogenea, for example). For this purpose, representative, rather than exhaustive, sampling of taxa is adequate. While broader sampling including early-branching lineages would be essential for reconstructing detailed evolutionary histories, assessing marker performance is not required in the context of our study.

Second, their selection and/or use of the outgroup taxa seems inappropriate given the availability of better alternatives (ie Bothrioplana, a well-sequenced representative of the closest known relative of the Neodermata) and their goal of evaluating the utility of the individual mt/rRNA single-gene markers, including their assessment of the relative genetic distances between taxa. The inclusion of an outgroup will inevitably affect branch lengths of the ingroup taxa. If the authors insisted on the use of an outgroup, I would probably suggest using some alternative approach, like the Evolutionary placement algorithm (doi.org/10.1093/sysbio/syr010) that has been implemented in RAxML. Even better, such an analysis might be better performed only including the ingroup taxa.

Reply: We appreciate the suggestion regarding outgroup selection. The choice of Macrostomum lignano and Schmidtea mediterranea as outgroups was based on their widespread use in previous studies addressing deep phylogenetic relationships in flatworms. While closer outgroups, such as Bothrioplana, have been recommended in some contexts, other studies have used more distantly related free-living flatworms (Perkins et al., 2010; Park et al., 2007) to avoid potential problems of branch attraction. Since our goal was to evaluate marker performance across different genes and methods, rather than produce a definitive phylogeny, we opted for outgroups that are well-established in the literature for comparative purposes. It is important to mention that the outgroup has two main functions: to provide orientation to the phylogenetic tree and to test the monophyly of the ingroup. In the context of our research, the selection of Platyhelminthes species less closely related to Neodermata fulfills the first function of the outgroup, as we only seek to provide orientation to the phylogenetic relationships obtained with the different data sets and treatments. The monophyly of Neodermata has already been tested in other studies and is not the objective of this research. Therefore, we believe that the use of M. lignano and S. mediterranea may be functional for the purposes of our comparisons. Nonetheless, we acknowledge that different outgrouping strategies may affect tree topology and branch lengths, and agree that future studies could explore how alternative placements, such as the use of Bothrioplana or ingroup rooting, impact results under this framework.

Third (and I confess this point might be rather subjective), I see their focus/topic of the paper somewhat exhausted and a thing of the past. Platyhelminth systematics have entered the era of phylogenomics a decade ago and the field seems now rather focused on determining novel (and genuine) phylogenetic signals from multiple genomic loci, identifying sources of phylogenetic noise and potential sources of systematic error in the data and tree inference methods. Here, two notoriously known genomic loci are being once again re-considered, both of which has been scrutinized number of times previously in a similar manner both independently and combined. The most recent paper on the topic I'm aware of was the one by Zhang et al (2024) Strong mitonuclear discordance in the phylogeny of Neodermata and evolutionary rates of Polyopisthocotylea. Int J Parasitol 54:213–223. But really, other numerous papers have already scrutinized the phylogenetic signal of mt vs nuclear rRNA (as well as within each locus) over the last twenty years both in the Neodermata and within individual major neodermatan groups. This point would be invalid if the manuscript brought some novel evidence (or data) but I don't see anything discussed in the text that I haven't seen before.

Additionally, I have been wondering about two other particular things but did not see a rationalization anywhere in the text: i) What was the particular motivation to analyse the notoriously known fast-evolving (ie mt protein-coding) genes at the nucleotide level when they have been shown to be saturated at significantly lower taxonomic levels previously in different groups of parasitic flatworms? ii) What was the motivation to employ two (largely substitute) maximum likelihood programs?

Reply: We agree that over the past decade, phylogenetic studies of Platyhelminthes using genomic data have increased, especially those exploring deep relationships. In this era of genomics, the number of published mitochondrial genomes has increased rapidly, and many of them have been used for phylogenetic hypotheses at different taxonomic scales. Multiple studies show that the use of mitochondrial genomes in deep phylogenetic hypotheses is not appropriate due to sources of phylogenetic noise such as high mutation rates of genes and third codon positions, rate heterogeneity between genes, among others. Despite this, in the absence of multilocus data from other sources (especially nuclear), they are still frequently used in taxonomic studies of parasitic Platyhelminthes. Because of this, the motivation for our research is precisely to conduct an updated assessment indicating that in parasitic Platyhelminthes, each mitochondrial gene provides different information and produces different topologies depending on the dataset used. This assessment can serve as a reference for future studies that intend to use mitochondrial genes or genomes in phylogenetic questions.

i)Phylogenetic analysis of sequences at the nucleotide and amino acid level allows us to identify aspects of the molecular evolution of genes, such as synonymous and non-synonymous nucleotide substitutions, which in turn can reveal the noise that could explain some of the topological variability observed in previous studies or in analyses with different datasets.

ii)Although both programs are based on maximum likelihood, they use different algorithms and implementations. We used them to evaluate the robustness of phylogenetic inferences in different machine learning frameworks. This was complemented by Bayesian inference, which relies on a different statistic.

Based on those facts, I suggest the authors to reconsider the paper.

Reviewer #3: This study investigates the topological variability of phylogenies of the major Neodermata clades derived from using different genetic and molecular datasets. While this is an interesting exercise, I have some objections that should be considered before publication.

My main criticism concerns the use of unilocus data for phylogenetic inference involving high-order clades. For over two decades, it has been known that unilocus sequences are quite unreliable for this purpose (e.g. https://doi.org/10.1371/journal.pbio.1000602).

Reply: We agree that single-locus markers have known limitations for resolving deep phylogenetic relationships. However, these markers continue to be used in parasitology due to the limited availability of genomic data for many taxa. The primary objective of our study was not to recommend single-locus analyses as a substitute for phylogenomic approaches, but rather to systematically evaluate their performance and the topological variability they produce when applied to the major lineages of Neodermata.

So some conclusions (see 290-291) are trivial.

Reply: We appreciate the recommendation. We've eliminated several of the potentially trivial conclusions.

I gather from the text that a possible reason could be to determine the genes useful for distinguishing between closely related species (92-93). However, this is totally outside the scope of the study and the subsequent discussion on the suitability of certain sequences to study intraspecific variation (354-369) is not directly backed by evidence from the study. Therefore, the authors should better explain the rationale and relevance for incorporating unilocus analyses in their study.

Reply. We have modified the last paragraph of the introduction and removed determining genes that contribute to phylogenetic differences from our objective to avoid disagreement about what is covered in the discussion. We limit the research objective to investigating the variability of phylogenetic tree topologies within the Neodermata using different mitochondrial genes and molecular datasets, given that our discussion did not fully address it. In addition, we modified the discussion paragraph, considering the reviewer's observation that the use of these genes is not supported by our analysis.

In a broader context, the anticipated relevance of the study is not sufficiently demonstrated in the Introduction. The study is largely based on reproducing an approach that was published over 10 years ago (Laumer and Giribert, 2014). Therefore, the current effort is not shown to be particularly innovative, and the authors should provide a clearer explanation of the advantages of their approach and its potential contribution to the state of the art. Most evidence presented appears to confirm previous work.

Reply. Thank you for your comment. We've made changes to the introduction to highlight the difference with the study by Laumer and Giribert (2014), and to emphasize that this is the first study to evaluate the phylogenetic topologies built from each mitochondrial gene at deep taxonomic levels.

Specific comments

The main goal of the study is to investigate topological variability in phylogenies resulting from the combination of different datasets. This is not adequately reflected in the title.

Reply: We have modified the title of the work to be more in line with the work.

Fig. 2 does not seem to add anything additional to what has been indicated in the text.

Reply: Figure 2 has been removed

252. “This analysis helped…” This sentence is not about a result, but an interpretation or explanation of results. Delete and elaborate in the discussion if required.

Reply: Line deleted.

---

## [Decision Letter · Decision Letter 1]

5 Sep 2025

PONE-D-25-05422R1Evaluating topological variability in Neodermata phylogenies using mitochondrial and ribosomal gene markersPLOS ONE

Dear Dr. Caña Bozada,

Thank you for submitting your manuscript to PLOS ONE. After careful consideration, we feel that it has merit but does not fully meet PLOS ONE’s publication criteria as it currently stands. Therefore, we invite you to submit a revised version of the manuscript that addresses the points raised during the review process.

Although it is my view that most of the concerns that reviewer #2 has raised can be classified as concerns about the scientific importance of your work (which is not a criterion PLOS ONE uses to assess the suitability of manuscripts), they have identified a concern about the article´s intelligibility which must be addressed before the work can be published in PLOS ONE. Specifically, the referee has highlighted that there is no mention, in the results or discussion, of the “silhouette widths” we are told you have calculated in your methods section. I am, therefore, requesting that you modify your manuscript to either exclude mention of these calculations in the methods or mention the measurements in the results and discussion (and expect a further round of peer-review).

Please also correct the following typographical errors:

Abstract (line 33): Please change “three software” to “three software packages”

And

Discussion (line 283): Please change “uur objective” to “our objective”.

We look forward to receiving your revised manuscript.

Kind regards,

James Lee Crainey, Ph.D.

Academic Editor

PLOS ONE

Journal Requirements:

Additional Editor Comments:

Thank you for your revised manuscript, as you will see, reviewer #2 still has considerable concerns about the suitability of your manuscript for publication in PLOS ONE.

Although it is my view that most of the concerns that reviewer #2 has raised can be classified as concerns about the scientific importance of your work (which is not a criteria PLOS ONE uses to assess the suitability of manuscripts), they have identified a concern about the article´s intelligibility which must be addressed before the work can be published in PLOS ONE. Specifically, the referee has highlighted that there is no mention, in the results or discussion, of the “silhouette widths” we are told you have calculated in your methods section. I am, therefore, requesting that you modify your manuscript to either exclude mention of these calculations in the methods or mention the measurements in the results and discussion (and expect a further round of peer-review).

Please also correct the following typographical errors:

Abstract (line 33): Please change “three software” to “three software packages”

And

Discussion (line 283): Please change “uur objective” to “our objective”.

Reviewers' comments:

Reviewer's Responses to Questions

**Comments to the Author**

1. If the authors have adequately addressed your comments raised in a previous round of review and you feel that this manuscript is now acceptable for publication, you may indicate that here to bypass the “Comments to the Author” section, enter your conflict of interest statement in the “Confidential to Editor” section, and submit your "Accept" recommendation.

Reviewer #1: All comments have been addressed

Reviewer #3: (No Response)

2. Is the manuscript technically sound, and do the data support the conclusions?

Reviewer #1: Yes

Reviewer #3: Yes

3. Has the statistical analysis been performed appropriately and rigorously? 

Reviewer #1: Yes

Reviewer #3: I Don't Know

4. Have the authors made all data underlying the findings in their manuscript fully available?

Reviewer #1: Yes

Reviewer #3: Yes

5. Is the manuscript presented in an intelligible fashion and written in standard English?

Reviewer #1: Yes

Reviewer #3: Yes

6. Review Comments to the Author

Reviewer #1: (No Response)

Reviewer #3: This is a revised version of the study on the topological variability of phylogenies of the major Neodermata clades based on different genetic and molecular datasets. While I appreciate the changes made, the manuscript is more focused than the first version, I still remain unconvinced about the significance of this contribution.

My main point of criticism concerned the use of unilocus data. The authors indicate that their purpose was to assess their performance and the variability of the topologies they produce. However, I remain unconvinced that documenting these limitations adds substantial value. The lack of sufficient resolution for resolving higher-order clades has already been extensively demonstrated in the literature. Given this well-established limitation and the increasing availability of genomic tools, their value for use in future work remains questionable. I would rather restrict the study to multilocus analyses. However, I still find that study is not particularly innovative as the focus of future work on higher-order phylogenies is expected to move increasingly toward large genomic datasets rather than on a limited number of genetic markers.

Specific comments

I cannot see the usefulness of the topological comparisons analyses (l. 162-173). Topologies were assigned to 10 clusters, which is a very high number and difficult to interpret biologically. In addition, the only distinctions explicitly considered were those between multilocus and unilocus topologies (Dimensions1), and among different sets of unilocus topologies (Dimension 2). Materials and Methods mentions the computation of silhouette widths (referred to as “global silhouettes” in the manuscripts) to assess the quality of clustering, but this metric is not shown in the results. How well separated the clusters were?

7. PLOS authors have the option to publish the peer review history of their article (what does this mean? ). If published, this will include your full peer review and any attached files.

**Do you want your identity to be public for this peer review?** For information about this choice, including consent withdrawal, please see our Privacy Policy .

Reviewer #1: No

Reviewer #3: No

---

## [Author Response · Author response to Decision Letter 2]

9 Sep 2025

Manuscript PONE-D-25-05422R1

Although it is my view that most of the concerns that reviewer #2 has raised can be classified as concerns about the scientific importance of your work (which is not a criteria PLOS ONE uses to assess the suitability of manuscripts), they have identified a concern about the article´s intelligibility which must be addressed before the work can be published in PLOS ONE. Specifically, the referee has highlighted that there is no mention, in the results or discussion, of the “silhouette widths” we are told you have calculated in your methods section. I am, therefore, requesting that you modify your manuscript to either exclude mention of these calculations in the methods or mention the measurements in the results and discussion (and expect a further round of peer-review).

Done. We have removed mention of silhouette width analysis and reference 35 which mentioned the package used.

Please also correct the following typographical errors:

Abstract (line 33): Please change “three software” to “three software packages”.

Done.

Discussion (line 283): Please change “uur objective” to “our objective”.

Done.

---

## [Editor Report · Decision Letter 2]

12 Sep 2025

Evaluating topological variability in Neodermata phylogenies using mitochondrial and ribosomal gene markers

PONE-D-25-05422R2

Dear Dr. Hernández-Mena,

We’re pleased to inform you that your manuscript has been judged scientifically suitable for publication and will be formally accepted for publication once it meets all outstanding technical requirements.

Kind regards,

James Lee Crainey, Ph.D.

Academic Editor

PLOS ONE

Additional Editor Comments (optional):

I am satisfied that the authors have made all my final requested corrections. It is, thus, now my opinion that there are no outstanding concerns preventing the author´s manuscript from meeting the PLOS ONE publication criteria. On this basis, I am happy to recommend this work for publication in PLOS ONE.
---

## [Editor Report · Acceptance letter]

PONE-D-25-05422R2

PLOS ONE

Dear Dr. Hernández-Mena,

I'm pleased to inform you that your manuscript has been deemed suitable for publication in PLOS ONE. Congratulations! Your manuscript is now being handed over to our production team.

Kind regards,

on behalf of

Dr. James Lee Crainey

Academic Editor

PLOS ONE